# Viral Loads in Ocular Fluids of Acute Retinal Necrosis Eyes Infected by Varicella-Zoster Virus Treated with Intravenous Acyclovir Treatment

**DOI:** 10.3390/jcm9041204

**Published:** 2020-04-22

**Authors:** Tomohito Sato, Wataru Yamamoto, Atsushi Tanaka, Haruna Shimazaki, Sunao Sugita, Toshikatsu Kaburaki, Masaru Takeuchi

**Affiliations:** 1Department of Ophthalmology, National Defense Medical College, Tokorozawa, Saitama 359-8513, Japan; dr21043@ndmc.ac.jp (T.S.); wtr.ym2t.mat@hotmail.co.jp (W.Y.); atanaka55@gmail.com (A.T.); lovemetender0714@gmail.com (H.S.); 2Laboratory for Retinal Regeneration, RIKEN Center for Biosystems Dynamics Research, Kobe, Hyogo 650-0047, Japan; sunaoph@cdb.riken.jp; 3Department of Ophthalmology, The University of Tokyo, Bunkyo, Tokyo 113-8654, Japan; kaburakito@gmail.com

**Keywords:** acute retinal necrosis, acyclovir, aqueous humor, detection rate, polymerase chain reaction, varicella-zoster virus, viral load, vitreous fluid

## Abstract

Acute retinal necrosis (ARN) is a rare viral endophthalmitis, and human herpesvirus is the principal pathogen. Early diagnosis and treatment are critical to avoid visual impairment by ARN, and pars plana vitrectomy (PPV) is required in advanced cases. In this study, we evaluated the transition of viral load in ocular fluids of ARN eyes with varicella-zoster virus (VZV) after intravenous acyclovir treatment. Fourteen eyes of 13 patients were analyzed retrospectively. All patients received intravenous acyclovir treatment, and eventually, all eyes underwent PPV. A polymerase chain reaction (PCR) test showed a 100% detection rate in all aqueous humor samples collected before the treatment (Pre-AH), as well as aqueous humor (Post-AH) and vitreous fluid samples (VF), collected during PPV conducted after the treatment. Within eight days or less of acyclovir treatment, viral loads both in AH and VF did not decrease significantly. Furthermore, the viral load of Pre-AH had a strong correlation with that of VH. These data suggest that in ARN eyes with VZV infection, the AH sample for the PCR test was reliable to confirm the pathogen. We propose that short-term treatment of intravenous acyclovir may be insufficient for reducing intraocular viral load, and the Pre-AH sample could be a predictor of viral activity in the eyes after acyclovir treatment.

## 1. Introduction

Acute retinal necrosis (ARN) is a rare infectious viral endophthalmitis that causes necrotizing retinitis and results in devastating visual outcome if not accurately diagnosed and treated [1]. At present, members of the human herpesvirus (HHV) family such as herpes simplex virus (HSV) type 1, type 2, and varicella-zoster virus (VZV) are recognized as the principal pathogens of ARN [2,3]. Furthermore, both immunocompetent and immunosuppressed individuals irrespective of gender and age may develop ARN caused by HHV infection [4,5,6].

Early diagnosis and treatment for ARN are of great importance to avoid irreversible tissue damage and visual impairment in the affected eyes [1]. Laboratory methods including tests for antibodies in serum and ocular fluids, viral culture, retinal biopsy, and immunocytochemistry could be helpful in the diagnosis of ARN, although routine use of these tests has been hampered by poor sensitivity or specificity, limited availability of these tests, and excessive risk for the patients [6].

Quantitative real-time polymerase chain reaction (qPCR) has greatly advanced the diagnosis of viral retinitis by detecting the viral deoxyribonucleic acid (DNA) and accurately estimating the number of viral genome copies in ocular fluids such as aqueous humor (AH) and vitreous fluid (VF) [7,8]. To date, numerous studies have reported the results of qPCR test using AH and VF samples for the detection and identification of pathogenic viruses in ARN eyes [1,6,9,10,11]. We previously developed a combination PCR system consisting of multiplex qualitative PCR and qPCR and showed the usefulness of this system to detect HHV genomes in ocular fluids of eyes with uveitis [12]. Regarding the detection rates of pathogenic viruses, the qPCR test of ocular fluids showed positive rates of 79–100% for HSV or VZV in cases of suspected ARN [6]. Therefore, performing diagnostic pars plana vitrectomy (PPV) to collect ocular specimens for PCR test has been accepted as an adjunctive diagnostic procedure to guide optimum treatment for ARN [13]. 

Early antiviral drug therapy, including the treatment of intravenous acyclovir or oral valacyclovir (a prodrug of acyclovir), is the standard treatment for ARN [6]. Previous studies reported that the durations to reach maximum drug concentrations after the initial dose of intravenous acyclovir and an oral dose of valacyclovir were approximately one hour and two hours, respectively [14,15]. On the other hand, the anti-herpetic activity of acyclovir is by inhibition of duplication of herpes-specified DNA polymerase [6]; in other words, the efficacy of acyclovir is not by eliminating cells infected by HHVs, although the proliferation of HHVs is inhibited. To the best of our knowledge, there are only a few studies reporting viral loads in ocular fluids of ARN eyes before and after antiviral drug therapy [10,16,17,18]. Currently, the kinetics of viral load in ARN eyes that require diagnostic and therapeutic PPV after initiation of systemic antiviral treatment is not fully known.

The purposes of our study were to (1) compare the detection rates of pathogenic viruses by the combination PCR system using AH samples collected before the initiation of intravenous acyclovir treatment (Pre-AH), and aqueous humor (Post-AH), and vitreous fluid samples (VF) collected during PPV after the acyclovir treatment in eyes with suspected ARN, and (2) evaluate the changes of viral load in AH and VF samples before and after the acyclovir treatment in ARN eyes infected by VZV.

## 2. Methods

### 2.1. Subjects

This retrospective observational study was performed at the National Defense Medical College Hospital and The University of Tokyo Hospital in Japan, which are tertiary centers with specialty outpatient clinics for uveitis. The study period was from May 1, 2007, to November 1, 2017, and a consecutive series of 18 eyes of 17 patients diagnosed with ARN were enrolled. The final pathogenic viruses in the eyes confirmed by the combination PCR system using ocular fluids were VZV (17 of 18 cases, 94.4%) and cytomegalovirus (CMV, 1 of 18 cases, 5.6%). In the enrolled cases, 14 patients were treated with intravenous acyclovir (1500 mg three times/day, daily), two patients were treated initially with oral valaciclovir (3000 mg three times/day, daily), and one patient whose disease was initially suspected as bacterial endophthalmitis was treated with intravenous vancomycin hydrochloride (1000 mg twice/day, one day) and ceftazidime (2000 mg twice/day, one day) before undergoing diagnostic and therapeutic PPV (Table 1). The study protocol was approved by the Ethics Committees of National Defense Medical College (No. 2021 and No. 2022) and The University of Tokyo (No. 11531), and the procedures conformed to the tenets of the Declaration of Helsinki. Informed consent was obtained from all patients in this study.

### 2.2. Collection of Aqueous Humor and Vitreous Fluid Samples

Pre-AH samples were collected from 11 eyes with suspected ARN (Table 1). VF samples were obtained from 14 eyes. Among them, paired samples of Pre-AH and VF were collected from seven eyes, and paired samples of Post-AH and VF were collected from three eyes. An aliquot of 0.1 mL of AH was aspirated using a 30 G needle. Non-diluted VF sample was collected during diagnostic and therapeutic PPV. The AH and VF samples were transferred into sterile tubes and stored at −80 °C until processing. No complication associated with the sampling of AH and VF occurred. 

In patients No. 1 to No. 4 (4 eyes of 17 patients, 22.2%), Post-AH and VF samples were not collected, because at the time of PPV, the pathogenic viruses had already been identified by the combination PCR system using Pre-AH sample. In patients No. 5 to No. 11, VF sampling was necessary, because at the time of PPV, the pathogenic viruses had not been identified by the PCR test using Pre-AH sample.

### 2.3. Comparisons of Virus Detection Rates and Viral Loads in Ocular Fluids

In this study, ARN eyes infected by CMV (No. 9) and ARN eyes caused by VZV infection treated with oral valaciclovir (No. 6 and No. 8) or antibacterial drugs (No. 15) were excluded from further analyses, in order to reduce the biases caused by medications and pathogenic viruses when comparing virus detection rates and viral loads. In those comparisons, we only examined the virus detection rates of the combination PCR system and evaluated the viral loads in ocular fluids in ARN eyes infected by VZV treated with intravenous acyclovir (14 eyes of 13 patients). 

### 2.4. Qualitative Multiplex Polymerase Chain Reaction

Genomic DNAs of HHVs in the AH and VF samples were measured using two independent PCR assays (the combination PCR system): (1) a qualitative multiplex PCR and (2) a qPCR [12]. DNA was extracted from the samples using an E21 virus minikit (Qiagen, Valencia, CA, USA) installed on a robotic workstation for automated purification of nucleic acids (BioRobot E21, Qiagen). The multiplex PCR was designed to qualitatively measure the genomic DNAs of eight human herpes viruses, including HSV type 1, HSV type 2, VZV, Epstein–Barr virus, CMV, HHV type 6, HHV type 7, and HHV type 8. The PCR was performed using a LightCycler (Roche, Switzerland). 

The primers of the viruses and the PCR conditions were described in previous reports [19,20,21,22,23,24,25]. Sequences of primers and probes for HSV type 1 to HHV type 8 are shown in Table 2. Specific primers for the viruses were used with AccuPrime Taq (Invitrogen, Carlsbad, CA, USA) and subjected to 40 cycles of PCR amplification. Hybridization probes were then mixed with the PCR products. When the genomic DNAs of HHVs were detected by the multiplex PCR, the samples were further analyzed by qPCR only for the HHVs. Representative results of a VZV-positive case (Patient No. 11) detected by the combination PCR system using the VF sample are shown in Figure 1.

### 2.5. Quantitative Real-Time Polymerase Chain Reaction

The qPCR was performed using AmpliTaq Gold and the Real-Time PCR 7300 system (ABI, Foster City, CA). The sequences of the primers and probes are shown in Table 3. The primers and PCR conditions have been reported previously [26,27,28,29,30,31,32]. All the reaction mixtures were subjected to 45 cycles of PCR amplification. The viral copy number in the sample was considered to be significant when more than 50 copies/tube (5.0 × 10^3^/mL) were observed [21]. To ensure that no contamination of the PCR preparation occurred, DNA amplification and analysis of the amplified products were carried out in separate laboratories. If a patient was found to be positive by only one of the PCR methods (for example, positive by the multiplex PCR and negative by the qPCR), the sample was assessed as PCR negative [21].

### 2.6. Statistical Analysis

Statistical analyses were performed using the statistic add-in software for Excel (BellCurve for Excel^®^, SSRI Co., Ltd., Tokyo, Japan, and XLSTAT^®^, Addinsoft company, Paris, France). Data are expressed as mean ± standard deviation (median). Two-tailed Fisher’s exact test (for *n* < 4) was used to compare independent categorical variables. Two-tailed Kruskal–Wallis test was used for nonparametric comparisons of multiple groups. One-tailed Mann–Whitney U test was used for the nonparametric comparison of unpaired groups. Two-tailed Spearman’s rank correlation was used to assess the nonparametric correlation of paired groups. A *p* level of less than 0.05 was considered to be statistically significant.

## 3. Results

### 3.1. Clinical Characteristics and Viral Loads of Ocular Fluids in ARN Patients with VZV Infection

The clinical characteristics and viral loads of ocular fluids in ARN patients with VZV infection (14 eyes of 13 patients) are summarized in Table 4. The ARN patients with VZV infection were divided into four groups according to the type of ocular fluid sample tested as follows: (1) Pre-AH sample only, (2) Pre-AH and VF samples, (3) VF sample only, and (4) Post-AH and VF samples. Regarding the classification, there was no overlap of patients among four groups. There was no significant difference in age, gender, and detection rates of pathogenic viruses by the combination PCR system. The detection rate was 100% in all four groups.

### 3.2. Comparisons of Viral Loads in Ocular Fluids before and after Initiation of Intravenous Acyclovir Treatment

The viral loads of VZV in Pre-AH, Post-AH, and VF calculated from the combination PCR system are summarized in Figure 2. There was no significant difference in the viral loads between Pre-AH group (Total), and the Post-AH group (Figure 2A). To examine the influence of duration of intravenous acyclovir treatment on viral load in AH, the Pre-AH patients were divided into two subgroups as follows: (1) patients who underwent PPV within three days after collection of Pre-AH (Pre-AH within three days), and (2) patients who underwent PPV more than three days after collection of Pre-AH (Pre-AH over three days). There was no significant difference in viral load between the groups of Post-AH and Pre-AH within three days (Figure 2B) or Pre-AH over three days (Figure 2C), or between the groups of Pre-AH within three days and Pre-AH over three days (Figure 2D).

To examine the influence of duration of intravenous acyclovir treatment on viral load in VF, the VF patients were classified into two subgroups as follows: (1) patients who underwent PPV within three days after the initiation of the treatment (VF within three days) and (2) patients who underwent PPV after more than three days of the treatment (VF over three days). There was no significant difference in viral load between the two groups (Figure 2E).

The clinical characteristics of the Post-AH group, the subgroups of Pre-AH and the VF group are summarized in Appendix A. There were no significant differences in age, gender, and laterality between the groups of Pre-AH (Total) and Post-AH, Pre-AH within three days and Post-AH, Pre-AH over three days and Post-AH, Pre-AH within three days and Pre-AH over three days, or VF within three days and VF over three days.

### 3.3. Comparisons and Correlations of Viral Loads in Ocular Fluids of the Same Patients before and after Initiation of Intravenous Acyclovir Treatment

The viral load of VZV in Pre-AH, Post-AH, and VF collected from the same patients before and after initiation of intravenous acyclovir treatment are shown in Figure 3. There was no significant difference in viral loads between the groups of Pre-AH and VF. On the other hand, the viral load in VF was significantly and markedly (approximately 45 times) higher than that in Post-AH (Figure 3B-(b)).

The correlation of viral loads between the groups of VF and Pre-AH or Post-AH is shown in Figure 3. The viral loads in Pre-AH samples correlated extremely strongly (*p* = 2.22 × 10^−16^) with those in VF samples (Figure 3A-(c)). On the other hand, there was no significant correlation between the viral loads in VF and Post-AH samples (Figure 3B-(c)).

The clinical characteristics of the paired sample groups with VF and Pre-AH or Post-AH are summarized in Appendix A.

## 4. Discussion

ARN was reported for the first time by Urayama et al. [33] in 1971 as a syndrome of acute panuveitis with retinal periarteritis. At present, ARN is recognized as a rare infectious viral uveitis syndrome that manifests as a form of necrotizing retinitis and may have devastating visual outcome if not accurately diagnosed and treated [1]. PCR test is a useful method for early diagnosis and treatment to identify pathogenic viruses in ARN eyes [6]. In the present study, we aimed to investigate the usefulness of AH as a sample of PCR test for suspected ARN eyes, and to evaluate the transition of viral loads in ocular fluids before and after initiation of systemic antiviral treatment in real-world clinical practice. We obtained several noteworthy findings as follows: (1) All the ocular fluid samples, including the Pre-AH and Post-AH samples yielded a high detection rate (100%) of pathogenic viruses by the combination PCR system, regardless of the differences in the type of ocular fluid, collection time, and systemic antiviral treatment. (2) In cases of ARN with VZV infection, the viral loads both in the AH and VF did not decrease over a short period after intravenous acyclovir treatment. (3) The viral load in the Pre-AH strongly correlated with that in the VF and may be a predictor of intraocular viral activity after short-term treatment of intravenous acyclovir.

Intravenous treatment with acyclovir for 10 days [5,34,35,36] and the use of oral antiviral drugs (e.g., famciclovir or valacyclovir) for five weeks to 24 weeks [37,38] has been accepted as a standard treatment for ARN patients in the acute phase. Acyclovir is an acyclic purine nucleoside analog that is converted to acyclovir monophosphate by virus-encoded thymidine kinase, which yields high concentrations of acyclovir triphosphate, inhibiting viral DNA synthesis through competitive inhibition of viral DNA polymerase [6,39]. Therefore, the therapeutic efficacy of acyclovir is not the elimination of infected cells or HHVs but the inhibition of HHV proliferation. In our study, the viral loads in AH and VF samples were not reduced over time after the initiation of intravenous acyclovir treatment in ARN eyes with VZV infection (Figure 2). In the literature, the changes in viral loads over time in ocular fluids of ARN eyes have been reported [11,18,40]. Bernheim et al. [18] demonstrated that the viral load in the AH of ARN eyes with VZV infection was constant during 27.8 ± 24.9 days after initiation of intravenous acyclovir treatment and intravitreal injections of antiviral drugs, suggesting that the conventional 10-day intravenous acyclovir treatment may be insufficient in patients who do not show a rapid response. In addition, Abe et al. [41] reported that the viral loads in AH and VF samples before systemic antivirus treatments correlated negatively with the final visual acuity in ARN eyes with VZV infection. Therefore, therapeutic strategies that effectively reduce intraocular viral load should be considered in ARN eyes with VZV infection.

Adjunctive treatments for ARN have been performed as follows: (1) early PPV with or without silicone oil before rhegmatogenous retinal detachment (RRD), (2) laser retinopexy around areas of necrosis to prevent RRD, (3) systemic and/or local corticosteroids, (4) systemic antiplatelet agents, and (5) intravitreal antiviral drugs [6]. In particular, several studies have reported the benefits of early PPV in term of reducing the onset risk of RRD [36,42,43], although the contradictory result was also reported [44]. As for the effects of vitrectomy, diagnostic and therapeutic PPV for suspected ARN has dual roles. First, larger volumes of vitreous fluids can be collected as diagnostic specimens for cell culture, cytology, PCR and other tests. Second, the procedure directly removes dense vitreous opacities containing substantial pathogenic viruses [10]. Furthermore, the PCR test using ocular fluid samples reliably and rapidly confirms causative viruses with a low rate of adverse events in cases of suspected ARN [6,45]. Therefore, we propose that early diagnostic and therapeutic PPV using microincision surgery with a wide-viewing system [46,47] combined with antivirus drug therapy would be the preferred minimally invasive approach to treat ARN eyes by excluding heavy pathogenic viruses, when clinical symptoms are unclear and response to antivirus drug therapy is inadequate.

Aqueous humor tapping is more easily performed and has a low risk of complications compared to vitreous fluid biopsy [10,18,48]. Previous studies have reported no significant difference between AH and VF samples in the detection rate of ARN-causing pathogenic viruses by PCR test [16,17]. On the other hand, questionnaire-based research conducted by the British Ophthalmological Surveillance Unit reported that vitreous fluid biopsy was significantly more likely to yield viral DNA on a PCR test compared with the aqueous humor tap (92.6% vs. 46.2%) in ARN cases [10]. In our study, both AH and VF samples yielded a 100% detection rate by the combination PCR system. As a possible reason for the contradictory result, since all the patients in our study finally underwent diagnostic and therapeutic PPV, they were presumably relatively severe cases having high viral loads in ocular fluids and hence were more easily detected, whereas the subjects in the previous study were reported as general ARN cases with incomplete samplings of AH (13 of 45 patients) and VF (27 of 45 patients) [10]. 

In our study, the causative virus was promptly confirmed by the combination PCR system using Pre-AH samples in four cases (22.2%) of suspected ARN, and we were able to initiate effective treatment with systemic antiviral drugs for VZV before performing therapeutic PPV (Table 1). Appropriate selection of effective antiviral drugs is essential because the efficacies of antiviral agents vary depending on the ARN-causing pathogenic viruses such as HSV type 1, HSV type 2, VZV and CMV [3,6]. Regarding the load of the pathogenic virus, our results using the paired samples showed that the viral load in the VF was significantly and approximately 45 times higher than that in the AH samples of ARN eyes with VZV infection after the initiation of intravenous acyclovir treatment (Figure 3). From the practical point of view, the sampling of AH for PCR test during PPV is feasible and safer than that of VF in ARN eyes because AH tapping is minimally invasive with very rare adverse events [18,48]. Therefore, we speculate that the AH sample collected at the time when ARN is suspected or during PPV, is an ideal sample for a PCR test to promptly decide the optimal antiviral treatment.

A PCR test using AH and VF samples is in principal used as a definitive diagnostic method to confirm pathogenic viruses of ARN [6,11,12,16,17,18,45]. Early diagnosis and treatment of ARN are crucial to avoid irreversible tissue damage and visual impairment [1]. To date, an important question remains unanswered: What should be the optimal cutoff viral load used in qPCR test of ocular fluids? In previous studies, the cutoff viral loads used in the qPCR test ranged from 3.8 × 10^5^ to higher than 1.0 × 10^7^ copies/ml in ARN eyes with HSV infection [40,49], and from 2.0 × 10^2^ to higher than 4.8 × 10^6^ copies/ml in ARN eyes with VZV infection [11,18]. In the present study, we set the cutoff value to be higher than 5.0 × 10^3^ copies/ml for the combination PCR system [12]. The cutoff value should be sufficiently high to exclude a false-positive result. Yet, ARN is a critically severe disease with a poor visual outcome, and therefore treatment should be initiated if a viral load suggesting presumptive ARN is detected. A previous study indicates that a viral load higher than 2.0 × 10^2^ copies/mL is clinically significant [18]. Therefore, the cutoff values of viral loads in qPCR tests vary widely [18]. In the future, large-scale multicenter studies are necessary to establish the optimal cutoff values of viral load for different pathogenic HHVs when qPCR test is used to analyze ocular fluids from ARN eyes. 

Currently, the PCR test is a reliable method to confirm the pathogens of ARN. However, the high detection sensitivity of PCR test could lead to a false-positive result in some cases. As for other diagnostic methods, Goldmann–Witmer coefficient (GWC) analysis that compares the level of antibody (IgG) production in the intraocular fluid to that in serum measured by enzyme-linked immune sorbent assay, is also used for diagnosing pathogens in infectious uveitis [50,51,52]. Abe et al. [50] showed that the detectability of viral pathogens by GWC analysis was equal to that by PCR test using AH and VF samples in retinitis with VZV infection. In addition, Takase at al [52] proposed diagnostic criteria based on six items of early-stage ocular findings (1a to 1f), five items of clinical courses (2a to 2e), and the result of PCR test or GWC analysis using intraocular fluids. In their study, diagnostic criteria using only ocular findings and clinical courses showed an extremely high rate of correct diagnosis with a sensitivity of 0.89 and specificity of 1.00, which was the same as that of the diagnostic criteria consisting of ocular findings, clinical courses and results of the virologic test [52]. Therefore, comprehensive diagnosis based on the results of GWC analysis and the diagnostic criteria will be required to ensure proper treatment for unclassifiable acute endophthalmitis, if the dissociation between clinical features and result of PCR test is suspected.

The present study has several limitations due to the retrospective design, which means that the treatments of eyes with suspected ARN were selected by individual physicians. The patients with suspected ARN studied were enrolled in only two core university hospitals with specialty outpatient clinics for uveitis, and there are some concerns as follows: (1) The protocols of systemic antiviral treatments and diagnostic and therapeutic PPV were not predetermined, even though the treatments and the PPV procedures for suspected RRD were performed according to the standard protocols for ARN [53]. (2) Compared with previous reports [1,5,9,12,16,17,36,54], the clinical characteristics of the enrolled patients were inclined towards patients with ARN caused by VZV infection (17 of 18 cases, 94.4%). (3) There was a small number of ARN cases with VZV infection. Therefore, the assessment of the efficacy of intravenous acyclovir treatment in reducing the viral load of ocular fluids was inadequate. (4) The viral load detected by qPCR test does not absolutely imply the presence of active infectious virus particles, because the amplified viral DNA sequences may represent remnants of viral DNA previously inhibited by antiviral drugs, and may also include nonproductive (abortive) particles [18]. (5) In this study, we only used the combination PCR system to confirm the pathogen in the ARN eyes studied. Other appropriate methods, such as GWC analysis [50,51,52], were not used for diagnosing the pathogens of ARN. (6) Other clinical parameters such as visual acuity, disease duration from onset to complete regression of ARN, the incidence of RRD [34,36,55], the involvement of the fellow eye [56,57], inflammatory activities in the anterior segment [58] and vitreous body [59], and sites of retinal necrosis [60] were not examined in our study, because the clinical findings of anterior and posterior segments were obscure due to the opacities in the anterior chamber and vitreous body. (7) The enrolled patients referred from other facilities to our university hospitals were presumably relatively severe cases. Therefore, it is conceivable that suspected ARN cases with mild clinical findings were potentially excluded in our study. 

The strengths of this study are as follows: (1) The efficacy of intravenous acyclovir treatment in reducing viral loads of ocular fluids in ARN eyes with VZV infection was examined under relatively low biases of medications and pathogenic viruses as well as therapeutic interventions because other adjunctive treatments including laser retinopexy, administrations of systemic corticosteroids, antiplatelet agents and intravitreal antiviral drugs were not used in all the enrolled patients. (2) All the enrolled patients underwent diagnostic and therapeutic PPV. Therefore, there was no selection bias depending on the severity of clinical diagnosis based on clinical features in the evaluation of the efficacy of intravenous acyclovir treatment on viral load in ocular fluids. (3) Treatments of eyes with suspected ARN were given under conditions of real-world clinical practice, and the viral loads in the eyes were obtained from the clinical data of eyes with suspected ARN receiving standard treatments in the real-world clinical setting.

## 5. Conclusions

In cases of suspected ARN, aqueous humor specimen for PCR test is a reliable and safe sample with low risk of adverse events, which aids in confirming pathogenic viruses and ruling out other masquerading ocular diseases. In ARN eyes with VZV infection requiring subsequent PPV, intravenous acyclovir treatment alone may be inadequate for reducing intraocular viral load in the short term. In cases of suspected ARN, we propose that early diagnostic and therapeutic PPV combined with antivirus drug therapy would be the preferred approach for direct removal of pathogenic viruses.

## Figures and Tables

**Figure 1 jcm-09-01204-f001:**
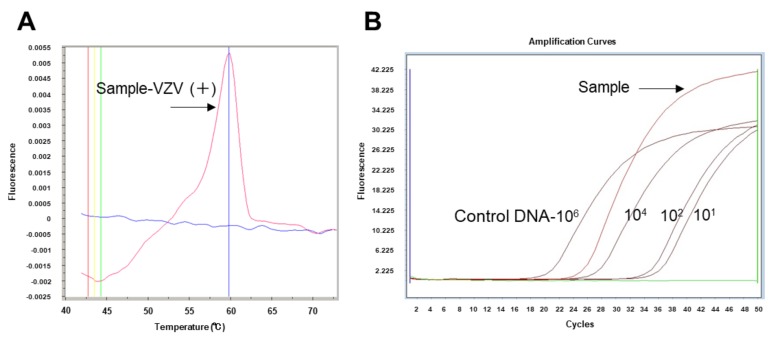
Representative results of a VZV-positive vitreous fluid sample (No. 11) analyzed using the combination PCR system. (**A**) Representative result of VZV-positive VF sample by qualitative multiplex PCR. After DNA extraction from the VF sample, multiplex PCR was performed to screen for viruses using LightCycler capillaries. At 60 °C, a significant positive curve was detected, indicating the detection of VZV-DNA in the sample. At the same time, other HHVs such as HSV type 1, type 2, and CMV were confirmed to be negative for this sample. (**B**) Representative result of the same VZV-positive VF sample analyzed by quantitative real-time PCR (qPCR). The VZV genome copy number in the sample was calculated. The VF sample and control DNA (1.0 × 10^6^, 10^4^, 10^2^, and 10^1^ copies/mL) by the qPCR were tested, and a standard curve using the results of control DNA was generated. Values were considered to be significant when more than 1.0 × 10^3^ copies/ml in the sample were observed. In the qPCR assay, we conducted with positive controls, negative control (water), and sample in the duplicate well assay. Representative single curves of each positive control and sample are present in the graph. PCR: polymerase chain reaction, VF: vitreous fluid, VZV: varicella-zoster virus.

**Figure 2 jcm-09-01204-f002:**
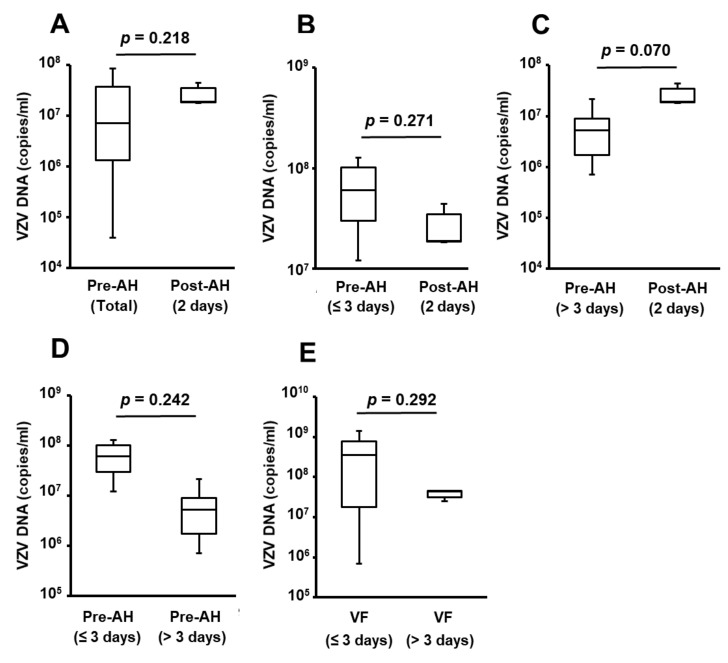
Comparisons of viral loads in ocular fluids before and after initiation of intravenous acyclovir treatment. Comparisons of viral loads between (**A**) Pre-AH (Total) and Post-AH (2 days), (**B**) Pre-AH (≤ 3 days) and Post-AH (2 days), (**C**) Pre-AH (> 3 days) and Post-AH (2 days), (**D**) Pre-AH (≤ 3 days) and Pre-AH (> 3 days) and (**E**) VF (≤ 3 days) and VF (> 3 days) are shown. The box plot represents the median, 25/75 percentiles, and 10/90 percentiles. Total: all of Pre-AH samples collected before PPV, 2 days: sample collected two days after initiation of the treatment, ≤ 3 days: sample collected within three days after initiation of the treatment, > 3 days: sample collected more than 3 days after initiation of the treatment. The number of samples in each group: Pre-AH (Total); eight eyes, Pre-AH (≤ 3 days); three eyes, Pre-AH (> 3 days); five eyes, Post-AH; three eyes, VF (≤ 3 days); seven eyes, VF (> 3 days); three eyes. Post-AH: aqueous humor collected during PPV conducted after the treatment. Pre-AH: aqueous humor samples collected before intravenous acyclovir treatment, PPV: pars plana vitrectomy, VF: vitreous fluid samples collected during PPV conducted after the treatment.

**Figure 3 jcm-09-01204-f003:**
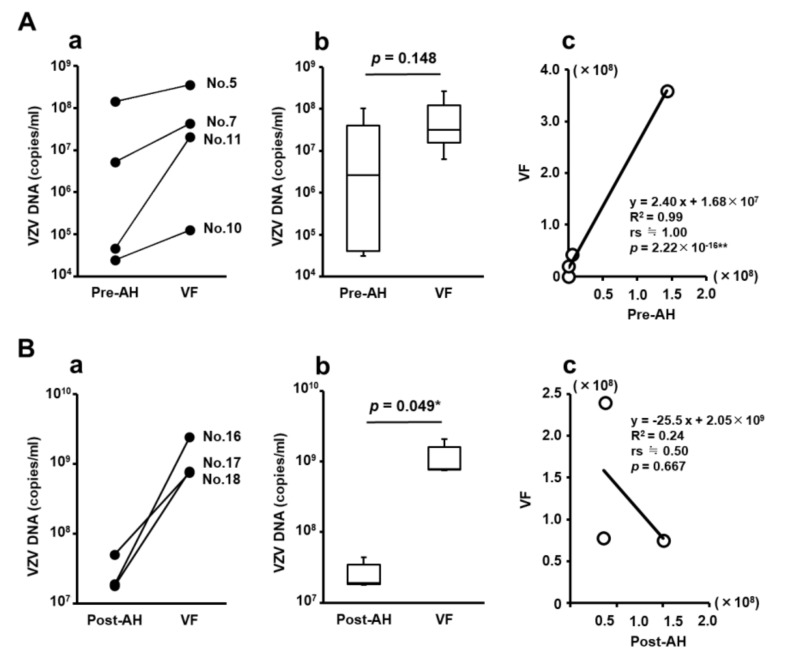
Comparison and correlation of viral loads in ocular fluids of the same patients before and after intravenous acyclovir treatment. (**A**) Paired samples of Pre-AH and VF in ARN patients with VZV infection. (**B**) Paired samples of Post-AH and VF in ARN patients with VZV infection. R^2^: coefficient of determination, rs: Spearman’s rank correlation coefficient, ≒: approximately equal, *: *p* < 0.05.

**Table 1 jcm-09-01204-t001:** Clinical summary of patients with suspected ARN and viral loads of ocular fluids.

Pt. No.	Age	Gender	Laterality	Final Diagnosis	Before Antivirus Treatment			Drug	Treatment Period	During Surgery					
					Sample	Pathogen	Viral Load			Sample	Pathogen	Viral Load	Sample	Pathogen	Viral Load
1	40	F	L	VZV	Aqueous humor	VZV	1.73 × 10^6^	Aciclovir i.v.	4 days						
2	45	M	R	VZV	Aqueous humor	VZV	8.87 × 10^6^	Aciclovir i.v.	6 days						
3	75	M	R	VZV	Aqueous humor	VZV	3.00 × 10^7^	Aciclovir i.v.	8 days						
4	50	M	R	VZV	Aqueous humor	VZV	6.00 × 10^7^	Aciclovir i.v.	3 days						
5	64	M	L	VZV	Aqueous humor	VZV	5.23 × 10^6^	Aciclovir i.v.	7 days				Vitreous fluid	VZV	4.30 × 10^7^
6	47	F	L	VZV	Aqueous humor	VZV	1.23 × 10^7^	Valaciclovir oral	1 day				Vitreous fluid	VZV	1.64 × 10^8^
7	81	M	L	VZV	Aqueous humor	VZV	1.43 × 10^8^	Aciclovir i.v.	1 day				Vitreous fluid	VZV	3.59 × 10^8^
8	53	M	R	VZV	Aqueous humor	VZV	1.53 × 10^8^	Valaciclovir oral	5 days				Vitreous fluid	VZV	1.03 × 10^9^
9	67	M	L	CMV	Aqueous humor	CMV	1.74 × 10^7^	Aciclovir i.v.	1 day				Vitreous fluid	CMV	6.10 × 10^7^
10	64	M	L	VZV	Aqueous humor	VZV	2.47 × 10^4^	Aciclovir i.v.	3 days				Vitreous fluid	VZV	1.26 × 10^5^
11	45	F	L	VZV	Aqueous humor	VZV	4.62 × 10^4^	Aciclovir i.v.	8 days				Vitreous fluid	VZV	2.05 × 10^7^
12	86	F	L	VZV				Aciclovir i.v.	1 day				Vitreous fluid	VZV	1.06 × 10^6^
13 *	75	F	R	VZV				Aciclovir i.v.	1 day				Vitreous fluid	VZV	3.50 × 10^7^
14 *	75	F	L	VZV				Aciclovir i.v.	5 days				Vitreous fluid	VZV	4.40 × 10^7^
15	83	F	L	VZV				Vancomycin/Ceftazidime i.v.	0 day				Vitreous fluid	VZV	2.41 × 10^8^
16	41	M	L	VZV				Aciclovir i.v.	2 day	Aqueous humor	VZV	1.89 × 10^7^	Vitreous fluid	VZV	2.40 × 10^9^
17	77	F	R	VZV				Aciclovir i.v.	2 day	Aqueous humor	VZV	1.80 × 10^7^	Vitreous fluid	VZV	7.80 × 10^8^
18	50	M	R	VZV				Aciclovir i.v.	2 day	Aqueous humor	VZV	5.02 × 10^7^	Vitreous fluid	VZV	7.47 × 10^8^

Eighteen eyes of 17 patients with suspected ARN were enrolled. The eyes of No. 13 and No 14 belonged to the same patient. In four cases (22.2%) of suspected ARN (upper rows with gray background), the pathogenic virus (VZV) was promptly confirmed by the combination PCR system using aqueous humor samples collected before systemic antiviral treatments, and optimal antiviral treatments were initiated before therapeutic PPV. Viral load is given in units of copies/mL. The PCR test was regarded positive in specimens with more than 5.0 × 10^3^ copies/mL, and the detected virus was confirmed as a pathogen. ARN: acute retinal necrosis, CMV: cytomegalovirus, i.v.: intravenous drip, PCR: polymerase chain reaction, PPV: pars plana vitrectomy, VZV: varicella-zoster virus, * Eyes of the same patient.

**Table 2 jcm-09-01204-t002:** Sequences of primers and probes for detecting human herpes viruses using qualitative multiplex PCR.

Herpes Virus	Primer Sequence	Probe Sequence	Amplication	References
HSV-1 and HSV-2 *	F: GCTCGAGTGCGAAAAAACGTTC	3′FITC: GCGCACCAGATCCACGCCCTTGATGAGC	polymerase	[19]
	R: TGCGGTTGATAAACGCGCAGT	LcRed604-5′: CTTGCCCCCGCAGATGACGCC		
varicella zoster virus	F: TGTCCTAGAGGAGGTTTTATCTG	3′FITC: GGGAAATCGAGAAACCACCCTATCCGAC	gene 29	[20]
	R: CATCGTCTGTAAAGACTTAACCAG	LcRed640-5′: AAGTTCGCGGTATAATTGTCAGT		
Epstein–Barr virus	F: CGCATAATGGCGGACCTAG	3′FITC: AAAGATAGCAGCAGCGCAGC	BamH1	[21]
	R: CAAACAAGCCCACTCCCC	LcRed640-5′: AACCATAGACCCGCTTCCTG		
cytomegalovirus	F: TACCCCTATCGCGTGTGTTC	3′FITC: TCGTCGTAGCTACGCTTACAT	CMV glycoprotein	[22]
	R: ATAGGAGGCGCCACGTATTC	LcRed705-5′: ACACCACTTATCTGCTGGGCAGC		
HHV type 6	F: ACCCGAGAGATGATTTTGCG	3′FITC: TAAGTAACCGTTTCGTCCCA	101K gene region	[23]
	R: GCAGAAGACAGCAGCGAGAT	LcRed705-5′: GGGTCATTTATGTTATAGA		
HHV type 7	F: GAAAAATCCGCCATAATAGC	3′FITC: GCCATAAGAAACAGGTACAGACATTGTCA	U57	[24]
	R: ATGGAACACCTATTAACGGC	LcRed705-5′: TTGTGAAATGTGTTGCG		
HHV type 8	F: AGCCGAAAGGATTCCACCAT	3′FITC: CCGGATGATGTAAATATGGCGGAAC	EB BDLF1 ORF26	[25]
	R: TCCGTGTTGTCTACGTCCAG	LcRed705-5′: TGATCTATATACCACCAATGTGTCATTTATG		

The qualitative multiplex PCR for HSV detects both HSV type 1 and type 2 DNA in the same reaction. * The optimized polymerase primer pair is common for HSV type 1 and type 2. On the other hand, the polymerase probe pair completely matches the genomic sequence of HSV type 2, but there are two base mismatches between the sequences of the probe pair and HSV type 1 genome. Therefore, the Tm of HSV type 1 is 15 °C higher than that of HSV type 2, and the difference in Tm provides a fine distinction between HSV type 1 and type 2. F: forward primer, HHV: human herpesvirus, HSV: herpes simplex virus, R: reverse primer.

**Table 3 jcm-09-01204-t003:** Sequences of primers and probes in human herpes viruses using quantitative real-time PCR.

Herpes Virus	Primer Sequence	Probe Sequence	Amplication	References
HSV type1	F: CGCATCAAGACCACCTCCTC	JOE-TGGCAACGCGGCCCAAC-TAMRA	gB	[26]
	R: GCTCGCACCACGCGA			
HSV type2	F: CGCATCAAGACCACCTCCTC	FAM-CGGCGATGCGCCCCAG-TAMRA	gB	
	R: GCTCGCACCACGCGA			
varicella zoster virus	F: AACTTTTACATCCAGCCTGGCG	FAM-TGTCTTTCACGGAGGCAAACACGT-TAMRA	ORF29	[27]
	R: GAAAACCCAAACCGTTCTCGAG			
Epstein–Barr virus	F: CGGAAGCCCTCTGGACTTC	FAM-TGTACACGCACGAGAAATGCGCC-TAMRA	BALF5	[28]
	R: CCCTGTTTATCCGATGGAATG			
cytomegalovirus	F: CATGAAGGTCTTTGCCCAGTAC	FAM-TGGCCCGTAGGTCATCCACACTAGG-TAMRA	IE-1	[29]
	R: GGCCAAAGTGTAGGCTACAATAG			
HHV type 6	F: GACAATCACATGCCTGGATAATG	FAM-AGCAGCTGGCGAAAAGTGCTGTGC-TAMRA	U65-U66	[30]
	R: TGTAAGCGTGTGGTAATGTACTAA			
HHV type 7	F: CGGAAGTCACTGGAGTAATGACAA	FAM-CTCGCAGATTGCTTGTTGGCCATG-TAMRA	U37	[31]
	R: CCAATCCTTCCGAAACCGAT			
HHV type 8	F: CCTCTGGTCCCCATTCATTG	FAM-CCGGCGTCAGACATTCTCACAACC-TAMRA	ORF65	[32]
	R: CGTTTCCGTCGTGGATGAG			

The quantitative real-time PCR for HSV is a multiplexing PCR that detects both HSV type 1 and type 2 DNA in the same reaction. The optimized gB primer pairs amplify both HSV type 1 and type 2 with equal efficiency, but the two type-specific probes are labeled with different fluorescent dyes. HSV type 1 probe is labeled with JOE at the 5′-end and with TAMRA at the 3′-end. HSV type 2 probe is labeled with FAM at the 5′-end and with TAMRA at the 3′-end. FAM: carboxyfluorescein, TAMRA: tetramethylrhodamine.

**Table 4 jcm-09-01204-t004:** Clinical characteristics of ARN patients with VZV infection classified into four groups according to type of ocular fluid sample.

Specimen	Pre-AH	Pre-AH and VF	VF	Post-AH and VF	*p* Value
*N*	4	4	3	3	Among 4 goups
Age (year)	52.5 ± 15.5 (47.5)	63.5 ± 14.7 (64.0)	80.5 ± 7.78 (80.5)	56.0 ± 18.7 (50.0)	0.277
Gender (M/F)	3/1	3/1	0/2	2/1	0.531
Laterality (R/L)	3/1	0/4	1/2	2/1	0.531
Detection *N* (%)	4 (100)	4 (100)	3 (100)	3 (100)	—

ARN patients with VZV infection were divided into four groups according to the type of ocular fluid sample. There was no overlap of patients among the four groups. Data are expressed as means ± standard deviations (median). Post-AH: aqueous humor samples collected during PPV after intravenous acyclovir treatment, Pre-AH: aqueous humor samples collected before the treatment, VF: vitreous fluid samples collected during PPV after the treatment. F: female, L: left, M: male, *N*: number, R: right.

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
