# Peer review of "Viral Loads in Ocular Fluids of Acute Retinal Necrosis Eyes Infected by Varicella-Zoster Virus Treated with Intravenous Acyclovir Treatment"

_jcm, 2020, doi:10.3390/jcm9041204_

Round 1

Reviewer 1 Report

I am satisfied with revised manuscript and recommend  for publication.

Author Response

Responses to the Comments of Reviewers

  We thank the editor and referees for taking their time to review our manuscript. Following Reviewer’s suggestions, we have added a new paragraph in Discussion about the usefulness of Goldmann-Witmer coefficient analysis and diagnostic criteria based on clinical features, complementing the result of PCR test for diagnosing pathogen of ARN. We have also added the limitation of diagnosis depending on the result of PCR test. All the coauthors have read and agreed with the changes made in the revised manuscript.

We hope that the corrections and revisions are satisfactory, and that the revised version will be acceptable for publication. We thank once again Reviewers for the constructive comments. Our responses to the comments and the changes are summarized below.

Responses to Reviewer #1

Comment 1

I am satisfied with revised manuscript and recommend for publication.

Response to Comment 1

We thank the Reviewer #1 for taking time to review our manuscript. We are really glad to receive a positive reply from the reviewer.

Reviewer 2 Report

The study by Sato et al. entitled ‘Viral Loads in Ocular Fluids of Acute Retinal Necrosis Eyes Infected by Varicella-Zoster Virus Treated with Intravenous Acyclovir’ compares the detection rates of pathogenic viruses by PCR using AH samples before acyclovir treatment (Pre-AH), and aqueous humor (Post-AH) and vitreous fluid samples (VF) collected during pars plana vitrectomy (PPV) after acyclovir treatment in eyes with suspected acute retinal necrosis (ARN), and (2) evaluated the changes of viral load in AH and VF samples before and after acyclovir treatment in ARN eyes infected by VZV.

I was amongst the reviewers of this article in the last submission. I had many comments and the majority of these comments have been answered by the authors, however, I have some remaining comments that need to be addressed in this round. But first of all, I would like to highlight the positive changes in this article.

In the present form, it sends a much clearer message, as it only deals with one viral infection except dealing with many with low sample numbers. The presentation of this research became more pleasant.

Now, I must address the limitations of the study. It relies only on PCR. There is only one technique in this study that had been used to produce the results even that is really sensitive. The use of protein-based assays could widen the range of information acquired. Also, as a clinical study, OCT or related techniques could add individual differences that could also of interest to the readers, but the effort made in giving an in-depth analysis really gives this paper a point to show.

My main concern is that I could not find the tables and figures in the PDF version of the article because they were completely missing, although I found the references in the text and found the supplements, therefore I will send back the paper for the authors as a major revision to address this issue and I am truly waiting for the updates to see the improved tables and figures.

Author Response

Responses to the Comments of Reviewers

We thank the editor and referees for taking their time to review our manuscript. Following Reviewer #2’s suggestions, we have added a new paragraph in Discussion about the usefulness of Goldmann-Witmer coefficient analysis and diagnostic criteria based on clinical features, complementing the result of PCR test for diagnosing pathogen of ARN. We have also added the limitation of diagnosis depending on the result of PCR test. All the coauthors have read and agreed with the changes made in the revised manuscript.

We hope that the corrections and revisions are satisfactory, and that the revised version will be acceptable for publication. We thank once again Reviewer #2 for the constructive comments. Our responses to the comments and the changes are summarized below.

Responses to Reviewer #2

Comment 1

Now, I must address the limitations of the study. It relies only on PCR. There is only one technique in this study that had been used to produce the results even that is really sensitive. The use of protein-based assays could widen the range of information acquired. Also, as a clinical study, OCT or related techniques could add individual differences that could also of interest to the readers, but the effort made in giving an in-depth analysis really gives this paper a point to show.

Response to Comment 1

We thank the reviewer for the constructive comments. We apologize for insufficient discussion about the detection sensitivity of the PCR test, and the availability of other tests.

In discussion section, we have added a paragraph describing the usefulness of Goldmann-Witmer coefficient analysis and diagnostic criteria based on clinical features, which would complement the result of the PCR test for diagnosing pathogens of ARN as follows:

Currently, the PCR test is a reliable method to confirm pathogens of ARN. However, the high detection sensitivity of PCR test could lead to false positive result in some cases. As for other diagnostic methods, Goldmann-Witmer coefficient (GWC) analysis that compares the level of antibody (IgG) production in intraocular fluid to that in serum measured by enzyme-linked immune sorbent assay, is also used for diagnosing pathogens in infectious uveitis [50-52]. Abe et al. [50] showed that the detectability of viral pathogens by GWC analysis was equal to that by PCR test using aqueous humor and vitreous fluid in varicella-zoster retinitis. In addition, Takase at al [52] proposed diagnostic criteria based on 6 items of early-stage ocular findings (1a to 1f), 5 items of clinical courses (2a to 2e) and the result of PCR test or GWC analysis using intraocular fluids. In their study, diagnostic criteria using only ocular findings and clinical courses showed extremely high rate of correct diagnosis with sensitivity of 0.89) and specificity of 1.00, which was the same as that for the diagnostic criteria consisting of ocular findings, clinical courses and results of virologic test [52]. Therefore, comprehensive diagnosis based on the result of GWC analysis and the diagnostic criteria will be required to ensure proper treatment for unclassifiable acute endophthalmitis, if dissociation between clinical features and PCR result is suspected.” (page 20-21, lines 324-340). Furthermore, we have added a limitation concerning diagnosis depending on the result of PCR test in our study as follows:

“(5) In this study, we only used the combination PCR system to confirm the pathogen in the ARN eyes studied. Other appropriate methods such as GWC analysis [50-52] were not used for diagnosing the pathogens of ARN.” (page 22, lines 356-358).

Comment 2

My main concern is that I could not find the tables and figures in the PDF version of the article because they were completely missing, although I found the references in the text and found the supplements, therefore I will send back the paper for the authors as a major revision to address this issue and I am truly waiting for the updates to see the improved tables and figures.

Response to Comment 2

We apologize sincerely for the missing tables and figures in the pdf of our first revised manuscript. We surely made the first revised manuscript including all of Tables and Figures as well as Supplemental Tables, and up-loaded it both in formats of PDF and Word. We also confirmed the first revised manuscript for peer review version. The manuscript for peer review version contains only the contents and do not include any Tables and Figures as well as Supplemental Tables. As for the concern, we guess that in the process of editing the first revised manuscript in JCM version, the Tables and Figures did not be inserted in the contents, unintentionally. We can view the Supplemental Tables of the first revised manuscript by clicking the button of Supplementary in the review-editing page.

When we submit the second revised manuscript, we will make sure to upload all the elements of the submission, including the manuscript, tables, figures and supplemental materials.

In addition, we will submit tables, figures and supplemental materials in supplemental section for the reviewer to confirm them.

In the second revised manuscript, all of tables, figures and Supplemental materials are the same in the first revised manuscript.

Round 2

Reviewer 2 Report

I accept the updates and answers of the authors. I have only two critical comments after seeing the figures.

In Figure 1. the axis labels are not visible. They are too small and the resolution of the figure is poor.

As I indicated in my first review the methodology in this manuscript is not detailed enough, even if it referenced older papers in methodology. My second comment arose from this fact. Were there technical replicates for the controls and samples in the qPCR?  As Fig 1b. shows the amplification curves one might suspect there were no replicates, meaning the controls and samples were measured in a single well each. In my opinion, this is not acceptable. If this was not the fact, please mention it in the figure legends (if eg. averaged curves are present in the figure).

Author Response

Responses to the Comments of Reviewers

We thank the editor and referee for taking their time to review our manuscript. Following Reviewer #2’s suggestions, we revised Figure 1 and added several sentences in the legend of Figure 1. All the coauthors have read and agreed with the changes made in the revised manuscript.

We hope that the corrections and revisions are satisfactory, and that the revised version will be acceptable for publication. We thank once again Reviewer #2 for the constructive comments. Our responses to the comments and the changes are summarized below.

Responses to Reviewer #2

Comment 1

In Figure 1. the axis labels are not visible. They are too small and the resolution of the figure is poor.

Response to Comment 1

We thank the reviewer for the constructive comments. We apologize for the illegible axis labels of Figure 1. We revised the axis labels of Figure 1 to be easily read.

Comment 2

My second comment arose from this fact. Were there technical replicates for the controls and samples in the qPCR?  As Fig 1b. shows the amplification curves one might suspect there were no replicates, meaning the controls and samples were measured in a single well each. In my opinion, this is not acceptable. If this was not the fact, please mention it in the figure legends (if eg. averaged curves are present in the figure).

 Response to Comment 2

 We thank the reviewer for the constructive comments. In the qPCR, we perform duplicate well assay in positive control (diluted 1.0×106, 104, 102 and 101 copies/ml), negative control (water) and the sample, respectively. The two curves in each positive controls and sample are almost the same. We tried to present the two curves in a graph, but the graph is too busy. Therefore, we present the representative single curve of each positive controls and sample in the graph. The following sentence has been added to the legend of Figure 1 as follows: “In the qPCR assay, we conducted with positive controls, negative control (water), and sample in duplicate well assay. Representative single curves of each positive controls and sample are present in the graph.” (page 41, lines 657-659).

This manuscript is a resubmission of an earlier submission. The following is a list of the peer review reports and author responses from that submission.

Round 1

Reviewer 1 Report

The purposes of our study are to  examine the detection rates of pathogenic viruses by  multiplex qualitative PCR test using aqueous humor samples collected before initiation of systemic  antiviral treatment, and aqueous humor and vitreous fluid samples collected during PPV after the  treatment, and (evaluate the transition of viral load in aqueous humor and vitreous fluid samples  before and after the initiation of systemic antiviral treatment. Author states that PCR testing showed  significant correlation between viral  loads in Pre-AH and VF. These data suggest that PCR test using AH sample is reliable to confirm pathogens of ARN, and that viral load in ocular fluid does not decrease over time after systemic  antiviral treatment in ARN eyes that undergo PPV.

Major concerns:

1.Similar study had been conducted  by D. Bernheim Bernheim, D et al. “Time profile of viral DNA in aqueous humor samples of patients treated for varicella-zoster virus acute retinal necrosis by use of quantitative real-time PCR.” Journal of clinical microbiology vol. 51,7 (2013): 2160-6. doi:10.1128/JCM.00294-13. Author should discuss more about the significance of current study since similar study had been done earlier.

2. The values of viral copy number in the samples were considered to be positive  when values more than 5.0 x 103 copies/mL were observed . The sensitivity of assay should be increased to detect lower copy number. please refer to above paper.

3. PCR and qRT-PCR primer sequence should be listed. It will be better to show representative DNA gel results.

Reviewer 2 Report

Brief summary

The article entitled ‘Usefulness of Aqueous Humor as Sample for Detecting Viral Load in Ocular Fluids of Acute Retinal Necrosis in Real-World Clinical Practice’ by Sato et. al. has evaluated the transition of viral load after systemic antiviral treatment in ocular fluids obtained from eyes that underwent pars plana vitrectomy (PPV) for acute retinal necrosis (ARN). The purposes of this research were to (1) examine the detection rates of pathogenic viruses by multiplex qualitative PCR test using aqueous humor samples collected before initiation of systemic antiviral treatment, and aqueous humor and vitreous fluid samples collected during PPV after the treatment, and (2) evaluate the transition of viral load in aqueous humor and vitreous fluid samples before and after the initiation of systemic antiviral treatment.

As highlighted by the authors this study has many limitations, those come from the design. Therefore, I do not see that adding CMV could really strengthen the message. The efficacy of acyclovir together with PPV is a great question to answer, that this study, in this form, fails to answer properly. With a better in-depth evaluation of this question, this article could be a useful addition to any journal. The figures should be re-tailored to better serve the delivery of the message in much better quality. Also, in the study, the small sample size could indicate limitations, but the results are not correctly supporting the purposes of the study and this originates not only from the retrospective design. The reader might have a feeling that the re-grouping of patients just serves the purpose of adding more variables to those that already had been introduced in the study by adding multiple antiviral drugs and multiple viruses. Furthermore, the technique of the mpxPCR and qPCR has been already described on aqueous humour and vitreous fluid with HHV and CMV (as referenced in Sugita 2008, ref. 10), decreasing the overall scientific novelty of the article.

In summary, however, there are many valuable points in the study,  it is not acceptable in this form. It needs to be rethought and updated by either adding more CMV and HSV patients in the study or by dismissing them and refocus on the VZV and acyclovir-treatment with PPV. I would encourage the authors to reevaluate the design accordingly and try to improve the overall quallity of the figures.

I will write my specific comments for each paragraph with my grammatical suggestions in the specific comments.

Specific comments

Title

The title is too long but still not descriptive enough. Something like, ‘Detection of varicella zoster and cytomegalovirus in ocular fluids in acute retinal necrosis’ would be more appropriate in this form, but after rethought a more clear version would be highly recommended.

Introduction

The introduction is very superficial. What are the present protocols for HSV, VZV, CMV detection? What is the sensitivity of these protocols? Are there similar PCR based assays (eg. CDC guides)? These are important questions in order to show the robustness of the methods/approach detailed in this study. Without these, one could not fit the study in the big picture. There are many points rather belonging here than in the discussion. 

L23. there ‘was’ a significant

L43 such as the test of

L47. the diagnosis

L48. the pathogenic virus genes - The whole genome is only detectable through sequencing, but it is not necessary to confirm the presence of the virus.

L51. the PCR test

L59. Please use SI (ml).

Methods

The subjects in this study are mixed. We could only assume similar outcomes from the same type of treatment. I would strongly recommend to leave out the non-acyclovir treated subjects (n=3 in contrast to n=14) from the study to avoid any false assumptions by reducing the variables. If the authors would stick to these a higher ‘n’ would be needed to avoid statistical pitfalls.

Table 1. bears more information to highlight. The detected viral loads are changing greatly. This difference should be marked (maybe on a log scale). Is this difference comes from the increased free viral particles after antiviral treatment?

In 2.3. PCR: Were the primers and the methodology for detection the same as in ref. 11? More details would be required.

L81. the National

Results 

There are overlapping text and lines in Table 1-3. Please, add group numbers on the top of the tables.

In 3.2 it would be good to mention the fold- changes in viral loads.

Figures 1 and 2 have really poor quality and the text is disturbingly small. These figures are also overcrowded. Please highlight any significance or relevant +/- correlation. As these figures serve as the backbone of the article they need to be edited in a way that better serves the understanding of the paper.  Regrouping and color-coding them would increase the paper’s overall value.

L147., 152. among the four groups

L150. ,(2) a group

L188. the median

Discussion

Many parts of the discussion belong to the Introduction as they serve as a background for this paper. 

As highlighted by the authors this study has many limitations, those come from the design. Again, I do not see that adding CMV could really strengthen the message. The efficacy of acyclovir together with PPV is a great question to answer, that this study, in this form, fails to answer properly.

L221. in the type of ocular fluid

L252. a contradictory result / more contradictory results were

L259. the PCR test

L265. has a lower risk

L271. a 100% detection

L282. in the vitreous fluid

L301. the elimination of

L318. in the anterior

L330. of the efficacy

L333. in a real-world